# Platelet-Rich Plasma in Maxillary Sinus Augmentation: Systematic Review

**DOI:** 10.3390/ma13030622

**Published:** 2020-01-30

**Authors:** Holmes Ortega-Mejia, Albert Estrugo-Devesa, Constanza Saka-Herrán, Raúl Ayuso-Montero, José López-López, Eugenio Velasco-Ortega

**Affiliations:** 1Faculty of Medicine and Health Sciences (Dentistry), University of Barcelona, 08970 Barcelona, Spain; holmes.ortegamejia@gmail.com (H.O.-M.); constanzasakah@gmail.com (C.S.-H.); 2Oral Health and Masticatory System Group (Bellvitge Biomedical Research Institute) IDIBELL, Department of Odontostomatology, Faculty of Medicine and Health Sciences (Dentistry), University of Barcelona, 08970 Barcelona, Spain; albertestrugodevesa@gmail.com (A.E.-D.); raulayuso@ub.edu (R.A.-M.); 3Service of the Medical-Surgical Area of Dentistry Hospital, University of Barcelona, 08970 Barcelona, Spain; 4Department of Stomatology, Faculty of Dentistry, University of Seville, 41013 Seville, Spain

**Keywords:** platelet-rich plasma, platelet-rich fibrin, bone substitutes, maxillary sinus augmentation, sinus floor augmentation

## Abstract

Background: Sinus augmentation can be performed with or without grafting biomaterials, and to date, there is no quality evidence regarding the augmentation of the sinus floor using only platelet concentrates, which can improve the healing period and enhance bone regeneration by stimulating angiogenesis and bone formation. The main objective of this paper was to assess the effect of the sole use of platelet concentrates in sinus augmentation in terms of newly formed bone, augmented bone height, and clinical outcomes and to assess the additional beneficial effects of platelet-rich fibrin (PRF) in combination with other grafting biomaterials. Methods: A systematic review was conducted following Preferred Reporting Items for Systematic reviews and Meta-Analyses (PRISMA) guidelines. Pooled analyses were performed with the Review Manager software. Results: For sinus elevation only using platelet concentrates, 11 studies met the inclusion criteria and were included for qualitative synthesis. Only one study was a clinical trial, which reported improved outcomes for the allograft group compared to the titanium-PRF (T-PRF) group. A total of 12 studies where PRF was used in addition to grafting biomaterials met eligibility criteria and were included in the review. Results from meta-analyses provided no additional beneficial effects of PRF in sinus augmentation in terms of bone height and percentage of soft tissue area. There was a statistically significant lower percentage of residual bone substitute material in the PRF (+) group compared to the PRF (−) group. The percentage of newly formed bone was slightly higher in the PRF (+) group, but this was not statistically significant. Conclusion: There is no robust evidence to make firm conclusions regarding the beneficial effects of the sole use of platelet concentrates in sinus augmentation. However, studies have shown favorable outcomes regarding implant survival, bone gain, and bone height. The use of PRF with other grafting biomaterials appears to provide no additional beneficial effects in sinus lift procedures, but they may improve the healing period and bone formation. Well-conducted randomized clinical trials (RCTs) are necessary to confirm the available results to provide recommendations for the clinical practice.

## 1. Introduction

### Rationale

Among other objectives, implant dentistry seeks to restore the loss of dental and supportive structures, either because of an aesthetic and/or functional problem for the patient [1]. For this reason, the concept of osseointegration, introduced by Branemark in the 1950s, and the use of dental implants for prosthetic purposes has become a widely spread treatment option [2].

One of the big challenges that arise from the deficiency of hard and soft tissues is represented in atrophic maxilla [3]. Implant placement may be compromised due to the pneumatisation of the maxillary sinus, with subsequent posterior alveolar bone loss and lack of bone quality and/or quantity [4]. Hence, several treatment options have been developed during the last 60 years, but it is since the 1980s that the augmentation of the maxillary sinus has become a surgical treatment option. Although not exempt from complications [5], it improves the condition of the implant recipient site in the upper-posterior zone [6,7,8,9,10,11,12].

There are different surgical approaches to elevate the Schneiderian membrane in order to achieve the vertical height of the alveolar process required for implant placement with an adequate length [13,14]. These techniques include the lateral window approach, which was first developed by Tatum [7] in the mid-1970s and later described by Boyne and James in 1980 [6]. Tatum first suggested the transalveolar technique for sinus floor elevation in 1986 [7], which was later modified by Summers [15] in 1994. This modified approach is called the osteotome technique, and it uses a set of osteotomes of increasing diameters intended to increase the density of the soft maxillary bone and create an upward fracture of the maxillary sinus floor.

The sinus membrane elevation can be performed with or without the use of grafting materials [16]. The non-use of grafting materials with an immediate placement of implants was initially introduced by Lundgren et al [17]; it requires a sufficient residual vertical alveolar bone height in the posterior zone of the maxilla to achieve an adequate primary stability of the inserted implant necessary to preserve the augmented Schneiderian membrane, thus allowing a clot formation around the exposed surface of the implant in the sinus cavity. This approach has been reported in several systematic reviews including, mostly, short-term descriptive studies [18,19,20].

On the other hand, there are different grafting materials commonly used to elevate the maxillary sinus floor, including autograft, allograft, xenograft, alloplast, and growth factors [21,22]. The selection of the ideal grafting material has been a matter of controversy over the years. However, due to its osteoinductive, osteogenic, and osteoconductive properties, autogenous bone graft is considered the gold standard [23,24,25]. Autologous bone graft has some inconveniences, the most important ones being its greater morbidity and unpredictable reabsorption [26,27,28]. This has led to an increase in the use of biological and synthetic bone substitutes in order to simplify the surgical procedure and to avoid the aforementioned complications. All these approaches are well documented in numerous systematic reviews and meta-analyses [29,30,31,32,33,34].

Platelet concentrates, as initially described by Whitman et al. [35] are blood-derived products obtained after the centrifugation of a blood sample. Different techniques have been developed to obtain a variety of preparations. To obtain platelet-rich plasma (PRP) and plasma rich in growth factors (PRGF), citrated blood is used during the centrifugation process to prevent coagulation, resulting in liquid PRP and PRGF. For its gel form, thrombin and/or calcium chloride is added to induce fibrin polymerization creating a low-density fibrin gel preparation. For platelet-rich fibrin (PRF), blood is collected without any anticoagulant, and it is immediately centrifugated, during which the natural coagulation process occurs [36,37]. The use of autologous platelet concentrates in the different fields of dentistry started in the 1980–1990s. Since then, platelet concentrates and growth factors have been used to enhance guided tissue regeneration in periodontology, in the regeneration of the pulpodentin complex, and in the guided restoration of bone loss in oral surgery [38].

To date, there is no high-quality evidence regarding the augmentation of the sinus floor using only PRP, PRGF, or PRF. The combination of PRP and autologous bone for sinus lift seems to provide no additional beneficial effects in terms of implant survival rate, implant stability, bone height, marginal bone level, bone density, laminar bone and tissue volume, bone graft resorption, angiogenesis, and soft tissue healing, compared to the use of only autologous bone, as reviewed by Strauss et al. in 167 patients [39]. Another review, which analysed 81 patients, reported a short-term improvement in densitometry values and new bone formation, but its clinical relevance is unclear [40]. The use of PRP in combination with Beta-Tricalcium Phosphate (β-TCP), which was studied in 35 patients [30], or xenografts, in 127 patients [41,42], appears to show no additional clinical benefits. The addition of PRF to deproteinized bovine bone mineral (DBBM) has also not shown further benefits [30,41,42]. However, in vivo animal studies have shown that the sole use of platelet concentrates in sinus augmentation achieves a mean height of newly formed bone of up to 3.6 mm [43,44], and recent clinical studies have reported that they could be a successful procedure in sinus augmentation on residual ridges <5 mm [45] and achieve a mean height of newly bone of up to 4 mm when implants are simultaneously placed [46,47].

## 2. Objectives

The main objective was to assess the effect of the sole use of platelet concentrates in maxillary sinus augmentation in terms of newly formed bone, augmented bone height, and clinical outcomes such as implant stability and implant survival. Furthermore, to evaluate the additional effects of PRF in combination with other graft materials in sinus lift relative to the outcomes mentioned above, we conducted a systematic review that answered the following structure question (PICO):P = Patients requiring unilateral or bilateral maxillary sinus augmentationI = Sole use of platelet concentrates / PRF + grafting materialsC = Grafting materials or nothingO = Newly formed bone, augmented bone height, implant stability and implant survival

## 3. Methods

### 3.1. Eligibility Criteria

The review included articles that met the following inclusion criteria:Patients in need of unilateral or bilateral sinus augmentation before implant dental placement.Randomized Clinical Trials (RCT), Controlled Clinical Trials (CCT), and comparative studies assessing histological, histomorphometric, clinical, and radiographic outcomes on the additional effects of PRF in sinus augmentation versus the non-use of PRF.Specified follow-up period.

As the available literature on the effects of the sole use of platelet concentrates in sinus augmentation is scarce in terms of evidence provided by RCT and CCT, we considered including retrospective studies, prospective studies, and case series. Case series including fewer than five patients were excluded. In vitro and in vivo (animal) studies were also excluded from this review.

### 3.2. Information Sources

The electronic search was conducted in MEDLINE (PuBMed) and the Cochrane Central Register for Controlled Trials. No restriction on date publication was applied and only articles published in English were considered. References of relevant studies selected for potential inclusion were also searched. The last search was performed on 9 December 2019.

### 3.3. Search Strategy

For the MEDLINE (PubMed) search, the following search terms were used: 

i- Sole use of platelet concentrates in sinus floor augmentation:

(“Platelet rich fibrin”(All Fields) OR “platelet rich plasma” (All Fields) OR “PRP” (All Fields) OR “PRF” (All Fields) OR “PRGF” (All Fields) AND “sinus floor” (All Fields))

ii- PRF + grafting biomaterials in sinus floor augmentation:

(“Sinus Floor Augmentation” (Mesh) OR “sinus lifting” (All Fields)) AND (“Platelet-Rich Fibrin” (Mesh) OR (platetet-rich (All Fields) AND (“plasma” (MeSH Terms) OR “plasma” (All Fields))) OR ((“plasma” (MeSH Terms) OR “plasma” (All Fields)) AND rich (All Fields) AND (“intercellular signaling peptides and proteins” (MeSH Terms) OR (“intercellular” (All Fields) AND “signaling” (All Fields) AND “peptides” (All Fields) AND “proteins” (All Fields)) OR “intercellular signaling peptides and proteins” (All Fields) OR (“growth” (All Fields) AND “factors” (All Fields)) OR “growth factors” (All Fields])))

### 3.4. Study Selection

Three authors independently reviewed the titles and abstracts of the references located in the databases. Articles were selected whenever they appeared to meet the inclusion criteria. The same two authors independently reviewed the studies selected in the initial screening for full-text revision and final selection. Disagreements were resolved by discussion. If no consensus between the two authors could be achieved, a third reviewer was consulted.

### 3.5. Data Collection Process and Items

Two reviewers extracted the following data from the selected studies: year of publication, country, design, study period, follow-up, number of patients, number of sinuses, sinus lift complications, implant surgery, number of implants, intervention, and comparison group, which were arranged in a table. Outcome measures were arranged in another table, which included: radiographic, histomorphometric, clinical and postoperative complications assessment. Two authors revised the data collection.

### 3.6. Outcomes and Summary Measures

Primary outcomes were the percentage of newly formed bone, percentage of residual-bone substitute material, and percentage of soft tissue area assessed by histomorphometric analyses. Furthermore, augmented bone height (mm) was assessed by radiographic evaluation and clinical outcomes in terms of implant stability (implant stability quotient), implant survival (%), and postoperative complications.

### 3.7. Risk of Bias in Individual Studies

For RCT, the authors critically appraised each study for potential risk of bias by the Cochrane Collaboration’s tool for assessing risk of bias. Studies were classified as “low risk”, “unclear risk”, and “high risk” of bias.

### 3.8. Synthesis of Results

Pooled analyses were performed using a random-effects model. Heterogeneity between studies was assessed by the I^2^ statistics. Heterogeneity was considered statistically significant for a p value <0.1. Heterogeneity was interpreted as recommended by the Cochrane Handbook: 0%–40% was considered unimportant, 30%–60% was considered as moderate heterogeneity, 50%–90% was considered as substantial heterogeneity, and 75%–100% was considered as considerable heterogeneity. Outcomes were combined using the Review Manager software, version 5.3 (Cochrane community, available in: https://community.cochrane.org/help/tools-and-software/revman-5/revman-5-download; accessed in 1 November 2019)

## 4. Results

### 4.1. Study Selection

The search among the literature for the sole use of platelet concentrates on maxillary sinus augmentation yielded 132 studies, which were screened for eligibility. After title and abstract evaluation, 18 studies were selected for full-text review. Of these, seven studies failed to meet inclusion criteria and were excluded from the review. Reasons for exclusion were in vivo (animal studies) [43,44,48], case series with <5 patients [49,50,51], and the use of a platelet concentrate in combination with a graft material [52]. Finally, 11 studies were included for qualitative synthesis (Figure 1A). For the use of PRF in combination with other grafting materials, the search yielded 74 articles. After duplicates were removed, 62 articles were screened for eligibility based on their title and abstract, from which 21 were selected for full-text review. Eleven studies were excluded, as they did not meet the inclusion criteria, nine studies were excluded because they lacked a comparison group [53,54,55,56,57,58,59,60,61], one was excluded because it was compared with patients treated for another reason [62], and another was excluded because the intervention was not of interest for this study [63]. Additionally, two articles were identified through other sources and were incorporated in the review. Finally, 12 studies were included for qualitative synthesis (Figure 1B).

### 4.2. Study Characteristics

#### 4.2.1. Sole Use of Platelet Concentrates in Sinus Floor Augmentation:

Among the 11 included studies, two were case series [58,60], three were retrospective studies [53,64,65], five were prospective studies [46,47,66,67,68], and one was an RCT [45]. The minimum follow-up was six months with a maximum of six years. In total, studies involved 301 patients requiring maxillary sinus elevation and 223 requiring sinuses lift procedures. Five studies did not report on the number of sinus elevations [46,53,64,65,66]. A total of 481 implants were placed, all with an immediate protocol. The majority of the studies used PRF as the sole grafting material. The descriptive characteristics of the studies are shown in Table 1. The only RCT [45] had a follow-up of nine months, involving 18 patients, from whom 10 were allocated to the intervention group (titanium-PRF) and the others were allocated to the control group (Cells & Tissue Bank Austria, Cells & Tissue Bank Austria (CTBA) allograft). There were 18 maxillary sinus elevations, with no reported complications and 37 implant placements in a delayed protocol (Table 2).

#### 4.2.2. PRF + Grafting Biomaterials in Sinus Floor Augmentation:

Among the 12 studies, two were retrospective studies [69,70], one was a CCT [71], and nine were RCTs [41,42,45,72,73,74,75,76,77], of which three had a split mouth design [41,42,74]. Follow-up ranged from one week to two years. Studies involved 246 patients in need of unilateral or bilateral sinus floor augmentation, and a total of 298 sinus lift procedures were performed, of which 149 were allocated to the intervention group (PRF + grafting biomaterial). A total of 498 implants were placed with a delayed protocol. Only one study allocated patients in an early, intermediate, or late protocol for implant placement [76]. The majority of the studies combined PRF with bovine bone mineral (Bio-Oss) [41,42,69,73,74,76,77], while two combined PRF with alloplastic grafts [71,72], two combined PRF with bone allograft [70,75], and one used titanium-PRF as the only grafting biomaterial [45]. Summary descriptive characteristics of included studies are shown in Table 2. 

### 4.3. Risk of Bias within Studies

From the 12 studies assessing the effect of PRF on sinus floor augmentation, nine were RCT. The risk of bias for these trials is presented in Table 3. Seven studies were considered as having a high risk of bias, mainly because the lack of blinding of participants and personnel [41,45,72,73,75,76,77]. These studies also did not specify their methods for allocation concealment, and some did not report the methods for random sequence generation, potentially introducing selection bias. One study was assessed as having an unclear risk of bias [74], and only one study was assessed as having a low risk of bias [42]. 

### 4.4. Results of Individual Studies

Summary data of the results of included studies for the sole use of platelet concentrates on sinus floor augmentation are presented in Table 1. Results of individual studies for the use of PRF in combination with other grafting materials on sinus lift are illustrated in Table 4. 

### 4.5. Synthesis of Results

#### 4.5.1. Sole Use of Platelet Concentrates in Sinus Floor Augmentation

i- Radiographic assessment:

Data on bone gain and/or bone height was available for nine studies. Anitua et al. [53] reported a mean bone height of 8.8 ± 1.4 mm and a mean bone gain of 4.3 ± 2.0 mm using plasma-rich in growth factors (PRGF) as the only graft material after three years of follow-up in an immediate protocol for short implant placement. Mean bone gain in the other studies varied between a minimum of 3.2–3.4 mm [66,67,68] and a maximum of 10 mm [58,60]. Kanayama et al. [46] reported a statistically significant mean bone gain in sandblasted acid-etched implants compared to hydroxyapatite implants, and Molemans et al. [67] reported a higher mean bone gain (5.4 ± 1.5 mm) with the lateral sinus floor elevation approach versus the transalveolar technique (3.4 ± 1.2 mm). Two studies additionally reported similar mean bone height values of 11.5 mm after six months of follow-up [47,65] (Table 1).

ii- Histomorphometric assessment:

Data on the percentage of newly formed bone was available for only one study [58], which reported a mean of 33 ± 5% after six months of follow-up (Table 1).

iii- Clinical assessment:

Data on implant survival rate was available in eight studies. Six studies reported an implant survival rate >95% after different periods of follow-up. The study by Simonpieri et al. [60], with the longest follow-up, reported an implant survival rate of 100% after six years. Aoki et al. [64] reported an 85.5% chance of implant survival after a mean follow-up of 3.5 years (range: 1–7 years). In subgroup analyses, the cumulative survival rates were 100% for the residual bone height ≥4 mm group and 69.6% for the residual bone height <4 mm group (p = 0.004) (Table 1).

#### 4.5.2. PRF + Grafting Biomaterials in Sinus Floor Augmentation

i- Radiographic assessment:

Data on radiographic outcomes was available for four studies. Results from Bolukbasi et al. [69] showed that during the study period (2 years), the PRF + Bio-Oss group had less change in the height of grafted sinus floor for implant compared to the Bio-Oss + Bio-Gide group (p = 0.02). Two years after the prosthetic loading, both groups had similar values for the mean change in the height of grafted sinus with a slightly higher value in the intervention group (4.49 ± 0.29 mm versus 4.14 ± 0.20 mm) (Table 4). Pichotano et al. [42] did not find differences in graft volume dimensional changes between the leukocyte-platelet rich fibrin (L-PRF) + Bio-Oss group and the control group after healing periods of four and eight months, respectively (Table 4). Similarly, Nizam et al. [41] did not find statistically significant differences in the augmented bone height between the test and control group after six months of sinus lift procedure. However, the results from Olgun et al. [45] showed that there was a statistical significant difference in the augmented bone height between the titanium-PRF (T-PRF) group (used as only grafting material) and the Cells & Tissue Bank Austria (CTBA) allograft, favoring the latter (11.7 ± 2.4 mm versus 19.9 ± 7.4 mm, respectively). Authors reported that the CTBA allograft group had 53% better volume, 86% better density, and 69% better height compared to the T-PRF group (Table 4). Our pooled analyses yielded no statistically significant difference in the mean augmented bone height between the combination of PRF + grafting biomaterial versus use of a graft material (Figure 2A).

ii- Histomorphometric assessment:

Ten studies provided data on histomorphometric assessment. Regarding the percentage of new bone formation, the majority of the studies reported a similar mean percentage of newly formed bone for both the test and control groups, without a statistically significant difference between them, showing no apparent additional effects of PRF when combined with other grafting biomaterial (Table 4). The percentage of new bone formation varied across studies between 17% and 45% in the PRF + grafting biomaterial group and 13% and 33% in the grafting biomaterial group. Only Pichotano et al. [42] demonstrated an increased percentage of newly formed bone in the L-PRF + Bio-Oss group compared to the Bio-Oss group (44.6 ± 13.9% versus 30.0 ± 8.4%, respectively). Zhang et al. [77] reported that the percentage of new bone formation in the PRF group was 1.4 times higher compared to the control group (Bio-Oss) (p = 0.14). Results from meta-analyses showed that the percentage of new bone formation was 1.98% higher in the PRF (+) group compared to the PRF (−) group, although these findings were not statistically significant and had moderate heterogeneity (Figure 2B).

Studies related to the percentage of residual-bone substitute material obtained similar results, with mean values between 3.6% and 33% for the PRF + grafting biomaterial group and 13.7% and 34% in the grafting biomaterial group (control group) (Table 4). However, Zhang et al. [77] reported a higher mean percentage of residual graft biomaterial in the control group (Bio-Oss) than in the test group (PRF + Bio-Oss) (28.5 ± 12.0% versus 19.2 ± 6.9%, respectively. Likewise, Pichotano et al. [42] reported a significantly higher percentage residual bone graft material in the Bio-Oss group (13.7 ± 9.9%) compared to the L-PRF + Bio-Oss group (3.6 ± 4.2%) (p = 0.01). The pooled analyses showed a statistically significant lower percentage of residual bone substitute material in the PRF (+) group, with 4.60% less residual bone graft material compared to the PRF (−) group (mean difference: −4.60, 95% CI −8.41, −0.78) with no significant heterogeneity (I^2^ = 38%) (Figure 2C).

Five studies reported outcomes regarding the soft tissue (fibrous) area in the maxillary sinus cavity. Three of them [42,69,72] showed a slightly higher increase in the amount of soft tissue component in the control group compared to the intervention group (PRF + grafting biomaterial), but differences were not statistically significant (Table 4). Meta-analyses results showed a higher percentage of soft tissue area in the control group than in the PRF (+) group (mean difference: −0.53, 95% CI −4.21, 3.16), but differences were not statistically significant (Figure 2D).

iii- Clinical assessment:

Data on implant survival and implant stability was available for three studies. Studies reported a 100% implant survival rate for both the intervention and control groups after one year of follow-up [41,42] and three years of follow-up [76]. Implant stability was assessed through the implant stability quotient (ISQ), and there were no significant differences between compared groups in terms of stability values [45,76]. Pichotano et al. [42] reported higher ISQ values in the control group (75.1 ± 5.7) compared to the L-PRF group (60.9 ± 9.3) immediately after implant placement but no statistically significant difference between groups at implant loading (p = 0.99) (Table 4).

Gurler et al. [75] showed higher healing index scores in the L-PRF + Minner-Oss group compared to that of the control group (Minner-Oss) on the seventh (4.2 ± 0.9 versus 3.6 ± 0.7) and 14^th^ (4.7 ± 0.4 versus 4.4 ± 0.5) day, but there was no statistical significance (Table 4).

iv- Patient outcomes:

Two studies provided data on patients’ outcomes. Gurler et al. [75] described gradual improvement in the L-PRF group compared to the control group regarding postoperative pain, swelling, sleeping, eating, phonetics, activities of daily living, and missed work days, but the differences were not statistically significant (p > 0.05). Del Fabbro et al. [73] stated that the use of PRGF combined with Bio-Oss resulted in a significant reduction of the perceived pain during the second and third postoperative day as compared to the control group (Bio-Oss). Patients in the test group also reported consistently less swelling, less hematoma, and less discomfort regarding chewing and speaking throughout the evaluation period (one week) (Table 4).

## 5. Discussion

### 5.1. Summary of Evidence

The existing evidence proved to be scarce in terms of the use of only platelet concentrates in maxillary sinus augmentation, with a lack of available studies that compare the use of only platelet concentrates as the grafting material versus other commonly used biomaterials, indicating that the clinical application of PRF in sinus lift procedures is relatively new. The majority of the studies included small cohorts of patients needing sinus floor augmentation with short-term follow-up (6–12 months). Efficacy of the sole use of platelet concentrates could not be assessed given the nature of the study design and lack of a control group. However, results from the studies showed that the use of PRF as the sole filling material during sinus lift with immediate implant placement is a reliable method that could lead to new bone formation and bone gain. The study by Olgun et al. [45] was the only RCT that compared the use of T-PRF versus CTBA allograft. Their results showed no differences in terms of new bone formation and implant stability, but radiographic assessment demonstrated that the CTBA allograft group had a statistical significant better bone volume (53%), bone density (86%), and height (69%) compared to the T-PRF group.

The evidence regarding the use of PRF in combination with other grafting biomaterials in maxillary sinus floor augmentation seems to provide no additional beneficial effects in terms of augmented bone height, percentage of soft tissue area, implant survival rate, and implant stability. A statistically significant difference revealed a lower percentage of residual bone substitute material favoring the PRF (+) group when compared to the use of grafting biomaterials alone (mean difference: −4.60, 95% CI −8.41, −0.78) with no significant heterogeneity (I^2^ = 38%). In addition, our pooled analyses showed a slightly higher percentage of new bone formation in the PRF group when compared to the use of grafting biomaterials alone. This advantage may be possible due to its osteoinductive properties and the enhanced revascularization process, which allows for improved healing periods of the bone tissue as well. In that context, Choukroun et al. [70] reported no differences in newly formed bone between the PRF + freeze-dried bone allograft (FDBA) protocol versus the FDBA alone protocol, after 4 and 8 months of healing period, respectively, concluding that the addition of PRF could reduce the healing time prior to implant placement [70].

Our results are consistent with other previous systematic reviews [40,78,79,80] and, to our knowledge, this is the second systematic review to provide pooled analyses. The previous systematic review [40] included five RCTs, so we broadened the search and included 12 studies, from which nine were RCTs. Included studies varied in their methodological quality, the majority of the included studies being assessed as high risk of bias. Only one study [42] was assessed as low risk of bias and reported a statistical high percentage of new bone formation in the L-PRF group compared to the control group (44.6 ± 16.9% versus 30.0 ± 8.45, respectively) and no differences between groups regarding graft volume dimensional changes, implant stability, and implant survival rate.

### 5.2. Limitations

Due to the lack of high-quality evidence, results must be interpreted with caution. Most of the included studies failed to specify their methods for allocation concealment, and some did not report the methods for random sequence generation, potentially introducing selection bias. Due to the nature of the intervention of interest, the majority of the studies lack any blinding of participants and personnel introducing, theoretically, performing bias. The degree to which the lack of blinding can affect the outcome assessment depends somewhat on the type of outcome measure. However, most of the studies included in this review assessed objective outcomes. Controlled clinical trials, retrospective, and prospective cohort studies have an inherent higher risk of bias. Most of them failed to control other variables that could influence the results. The majority of the available studies are heterogeneous in terms of intervention, comparison, and outcome measures, hindering direct comparisons on a much larger scale and precluding assessing a precise estimate of the beneficial effects of PRF. Another limitation is the language restriction, as only studies published in English were included in this review.

### 5.3. Innovation and Challenges

Traditionally, autogenous bone has been used in sinus floor augmentation to increase the bone height needed for implant placement. Although it is a predictable and effective technique, it has a high morbidity rate and takes approximately six months to be integrated and substituted by osteoconduction [70], delaying the implant placement and, consequently, extending patient treatment. Autologous graft materials partially account for these inconveniences, avoiding the morbidity associated with the use of autogenous bone. The use of PRF clots is a simple, low-cost, and low-resource technique that could be used as a treatment option for sinus lift procedures, improving the healing period, regeneration, and maturation of bone tissue and thus shortening the implant placement stage. In addition to its osteoinductive property, PRF promotes angiogenesis reduces tissue inflammation and infectious reactions [75], and promotes an immune regulation through the gradual release of anti-inflammatory cytokines [81]. These properties and advantages made PRF a promising biomaterial for clinicians and maxillofacial surgeons in different fields of dentistry. Clinical applications of PRF in sinus lift augmentation have recently been investigated. Moreover, there is a lack of evidence provided by RCTs related to the effects of PRF as the sole grafting material in sinus lift procedures with atrophic maxillaries (<5 mm). Preliminary studies have described positive results in one-stage implant placement protocols regarding bone gain and implant survival, but it is necessary to compare the use of only PRF with the use of other biomaterials or the use of PRF in addition to other biomaterials to assess its effectiveness. 

## 6. Conclusions

There is no sufficient robust evidence to make firm conclusions regarding the beneficial effects of the sole use of platelet concentrates in sinus augmentation procedures. Although case series and cohort studies have shown favorable outcomes regarding implant survival, bone gain, and bone height, RCTs are required to assess whether platelet concentrates used as the only grafting materials are at least not inferior to autologous bone or other biomaterials used in sinus floor augmentation. The addition of PRF to other biomaterials appears to provide no further beneficial effects or to improve the outcomes in sinus lift procedures. Due to their biological properties, it could be considered as a reliable treatment option with the advantages of an improved healing period with a consequent earlier implant placement stage, enhanced new bone formation, and improved postoperative complications. Nevertheless, well-designed and conducted RCTs with the same PRF protocol preparation, blinding of personnel in charge of outcome assessment, and long follow-up periods are necessary to confirm the available results, determining their clinical implications and recommendations for the clinical practice.

## Figures and Tables

**Figure 1 materials-13-00622-f001:**
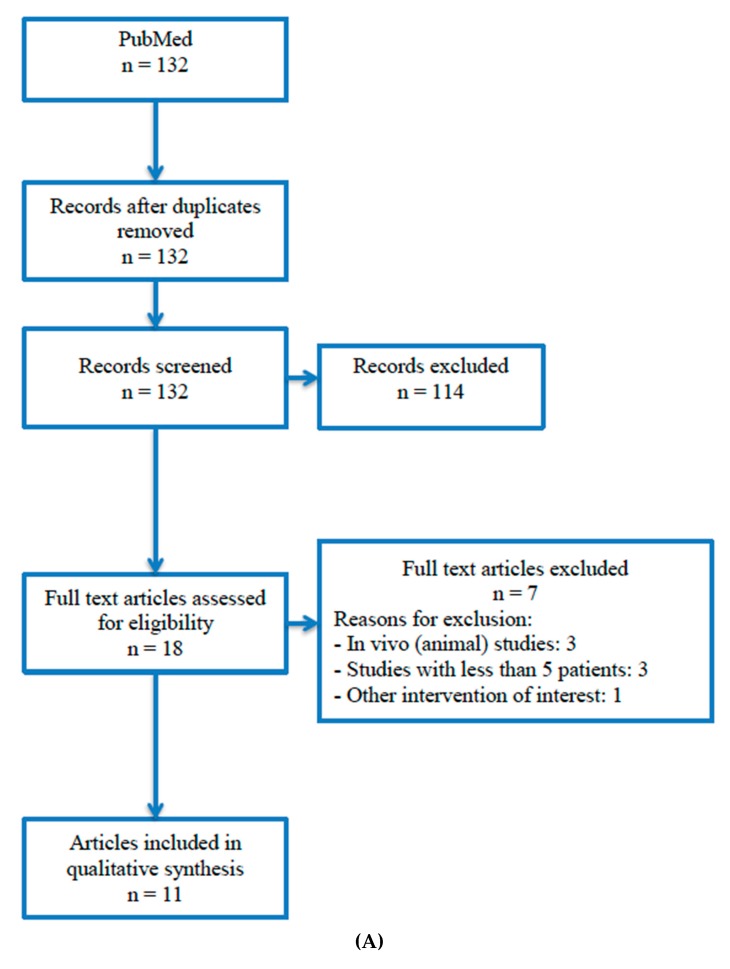
**(A)** Diagram for study selection: sole use of platelet concentrates in sinus floor augmentation. (**B)** Diagram for study selection: platelet-rich fibrin (PRF) in combination with other grafting biomaterials in sinus floor augmentation.

**Figure 2 materials-13-00622-f002:**
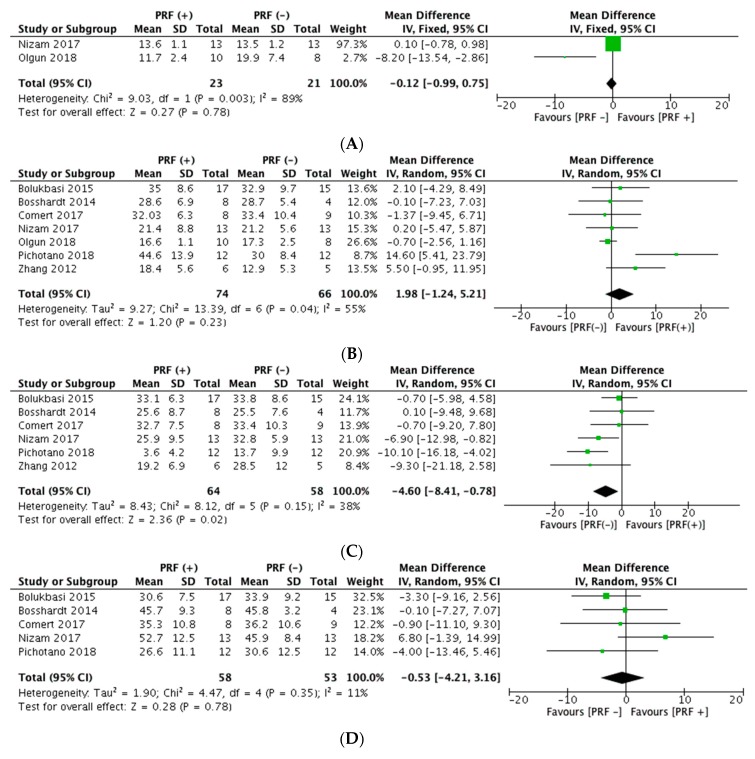
Plot for comparison: PRF and grafting biomaterial in sinus floor augmentation. (**A**) Augmented bone height (mm), (**B**) Percentage of new bone formation, (**C**) Percentage of residual bone-substitute material, and (**D**) Percentage of soft-tissue area.

**Table 1 materials-13-00622-t001:** Descriptive characteristics of studies included for sole use of platelet concentrates in sinus floor augmentation.

Author, Year Country	Design Study Period Follow-Up	Nº Patients	Gender Age	Nº Sinus	Sinus Lift Complications	Implant Surgical Stage	Nº Implants	Platelet Concentrate	Outcome Measures
Anitua et al, 2016 Spain [53]	Retrospective Study 3 years	26	F: 14 M: 12 55 ± 7y	NR	NR	Immediate placement	41	PRGF PRGF + bone graft in 7 implants	Radiographic assessment (mean ±sd) * Bone height (1year): 8.2 ± 1.2 mm Bone height (3 years): 8.8 ± 1.4 mm Bone gain (1 year): 3.6 ± 1.8 mm Bone gain (3 years): 4.3 ± 2.0mm
Aoki et al, 2018 Japan [64]	Retrospective Study 2010–2015 Χ = 3.43 years (1–7 y)	34	F: 17 M: 17 57.6y	NR	NR	Immediate placement	71	PRF	Clinical assessment Implant survival: 85.5%
Diss et al, 2008 France [66]	Prospective Study 2004–2005 1 year	20	F: 14 M: 6 54.8 ± 11.1y	NR	4 (Sinus membrane perforation)	Immediate placement	35	PRF	Radiographic assessment (mean ±sd) Bone gain: 3.2 ± 1.5mm (0.1–5.8 mm) Clinical assessment Implant survival: 97.1%
Gulsen et al, 2019 Turkey [65]	Retrospective Study 2015–2018 6 months	12	F: 7 M: 5 55.7 ± 8.3	NR	1 (Sinus membrane perforation)	Immediate placement	18	i-PRF soaked collagen	Radiographic assessment (mean ±sd) Bone height: 11.6 ± 1.6mm (8–14 mm) Bone gain: 6.3 ± 1.3mm (4.2–8.5 mm) Clinical assessment Implant survival: 100%
Kanayama et al, 2016 Japan [46]	Prospective Study 2011–2013 1 year	27	F: 15 M: 12 54.2y (29-74y)	NR	0	Immediate placement	39	PRF	Radiographic assessment (mean ±sd) Bone gain: SA implant: 4.4 ± 1.7 mm (p < 0.001) HA implant: 4.0 ± 1.6 mm Clinical assessment Implant survival: 100%
Mazor et al, 2009 Israel [58]	Case series 2007–2008 6 months	20	F: 14 M: 6 54.1 ± 5.2y	25	0	Immediate placement	41	L-PRF	Radiographic assessment (mean ±sd) Bone gain: 10.1 ± 0.9 mm (7–13 mm) Histomorphometric assessment (mean ±sd) Newly formed bone: 33% ± 5% Clinical assessment Implant survival: 100%
Molemans et al, 2019 Belgium [67]	Prospective Study 2015–2016 6 months	26	F: 12 M: 14 55y (38–78)	28	NR	Immediate placement	29	L-PRF	Radiographic assessment (mean ±sd) Bone gain: TSFE: 3.4 ± 1.2 mm // LSFE: 5.4 ± 1.5 mm Clinical assessment Implant survival: 93.1% (2 implants were not osseointegrated and were removed)
Simonpieri et al, 2011 France [60]	Case series 2003–2008 6 years	20	F: 12 M: 8 59.8 ± 11.1y	23	3 (Sinus membrane perforation)	Immediate placement	52	L-PRF	Radiographic assessment (mean ±sd) Bone gain: 10.4 ± 1.2 mm (8.5–12 mm) Clinical assessment Implant survival: 100%
Tajima et al, 2013 Japan [47]	Prospective Study 2009–2011 6 months	6	F: 6 67.8y (53–82y)	9	0	Immediate placement	17	PRF	Radiographic assessment (mean ±sd) Bone height: 11.8 ± 1.7 mm (9.1–14.1 mm) Bone gain: 7.5mm Bone density: 323 ± 156.2 HU (185–713) Bone volume: 0.7 ± 0.3 mL
Toffler et al, 2010 USA [68]	Prospective Study 2008–2010 11 months	110	F: 70 M: 40 58.4 y (34–90 y)	138	5 (Sinus membrane perforation)	Immediate placement	138	PRF	Radiographic assessment (mean ±sd) Bone gain: 3.4 mm (3.5–5 mm) Clinical assessment Implant survival: 96.4%

Abbreviations = F: Female; M: Male; y: year; PRGF: Plasma rich in growth factors; NR: not reported; i-PRF: injectable platelet-rich fibrin; HA: hydroxyapatite; SA: sandblasted acid-etched; L-PRF: leukocyte-platelet rich fibrin; TSFE: transalveolar sinus floor elevation; LSFE: lateral window sinus floor elevation. * Results for sole use of PRGF (34 implants).

**Table 2 materials-13-00622-t002:** Descriptive characteristics of studies included for the use of PRF combined with other grafting materials in sinus floor augmentation.

Author, Year Country	Design/ Study Period Follow Up	Nº Patients	Nº Sinuses	Sinus Lift Complications	Implant Surgery	Nº Implants	Intervention Group (I)	Control Group (C)
Bolukbasi et al, 2015 Turkey [69]	Retrospective Study 2008–2012 2 years	25	I: 17 C: 15	I: 0 C: 0	6 months after sinus lifting	I: 34 C: 32	Bovine bone graft material (Bio-Oss) + PRF mixture	Bovine bone graft material (Bio-Oss) + collagen membrane (Bio-Gide)
Bosshardt et al, 2014 Switzerland [71]	Controlled Trial 7–11 months	8 I: 5 C: 3	12 I: 8 C: 4	I: 0 C: 0	7-11 months after sinus lifting	16	Synthetic nanocrystalline hydroxyapatite embedded in highly porous silica gel matrix (NanoBone) + PRF	Synthetic nanocrystalline hydroxyapatite embedded in highly porous silica gel matrix (NanoBone) + collagen membrane (BioGide)
Choukroun et al, 2006 France [70]	Retrospective study 2001–2003 8 months	9	I: 6 C: 3	I: 1 (Perforation of the sinus membrane) C: 0	I: 4 months after sinus lifting C: 8 months after sinus lifting	NR	Freeze-dried bone allograft + PRF	Freeze-dried bone allograft
Comert et al, 2017 Turkey [72]	RCT 2012–2013 6 months	26 Group A: 9 Group B: 8 Control: 9	NR	Group A: 1 Group B: 2 Control: 2 (Perforation of the sinus membrane)	6 months after sinus lifting	NR	Group A: P-PRP mixed β- TCP Group B: PRF mixed β- TCP	β- TCP
Del Fabbro et al 2013, Italy [73]	RCT 2011–2012 1 week	30 I: 15 C: 15	NR	I: 1 C: 2 (Perforation of the sinus membrane)	6-8 months after sinus lifting	NR	Deproteinized bovine bone matrix (Bio-Oss) + PRGF-Endoret	Deproteinized bovine bone matrix (Bio-Oss)
Gassling et al, 2013 Germany [74]	Split mouth RCT 2010–2011 5 months	6	I: 6 C: 6	NR	5 months after sinus lifting	32	Autologous bone and bone-substitute material (Bio-Oss) mixed in 1:1 ratio + PRF	Autologous bone and bone-substitute material (Bio-Oss) mixed in 1:1 ratio + collagen membrane (Bio-Gide)
Gurler et al. 2016 Turkey [75]	RCT 2 weeks	28 (4 excluded from analyses) I: 12 C: 12	28 I: 12 C: 12	4 (2 postoperative maxillary sinusitis and 2 perforation of the sinus membrane)	NR	NR	Allogenous freeze dried corticocancellous bone chips (MinerOss) + L-PRF	Allogenous freeze dried corticocancellous bone chips (MinerOss)
Nizam et al, 2017 Turkey [41]	Split mouth RCT 2013-2015 6 months	13	26 I: 13 C: 13	I: 0 C: 0	6 months after sinus lifting	I: 30 C: 28	Deproteinized bovine bone mineral (BioOss) + L-PRF	Deproteinized bovine bone mineral (BioOss)
Olgun et al, 2018 Turkey [45]	RCT 2013–2014 9 months	18	18 I: 10 C: 8	I: 0 C: 0	I: 4 months after sinus lifting C: 6 months after sinus lifting	37	Titanium-PRF	Allograft (CTBA Allograft)
Pichotano et al, 2018 Brazil [42]	Split mouth RCT 2014–2015 8 months	12	22 I: 12 C: 12	I: 0 C: 0	I: 4 months after sinus lifting C: 8 months after sinus lifting	I: 19 C: 19	Demineralized bovine bone mineral (BioOss) + L-PRF	Demineralized bovine bone mineral (BioOss)
Tatullo et al, 2012 Italy [76]	RCT 150 days	60 I: 30 C: 18 Split mouth: 12	72	I: 0 C: 0	Early (106 days after sinus lift): 20 Intermediate (120 days after sinus lift): 20 Late (150 days after sinus lift): 20	240	Deproteinized bovine bone mineral (BioOss) + PRF	Deproteinized bovine bone mineral (BioOss)
Zhang et al, 2012 China [77]	RCT 6 months	11	11 I: 6 C: 5	NR	6 months after sinus lifting	I: 6 C: 5	Deproteinized bovine bone mineral (BioOss) + PRF	Deproteinized bovine bone mineral (BioOss)

Abbreviations: I: Intervention group; C: Control group; PRF: Platelet-Rich Fibrin; NR: Not Reported; RCT: Randomized Clinical Trial; P-PRP: Platelet-Rich Plasma; β- TCP: Beta-Tricalcium Phosphate; L-PRF: Leukocyte-PRF; PRGF: Plasma Rich in Growth Factors.

**Table 3 materials-13-00622-t003:** Risk of Bias of included randomized clinical trials.

Randomized Clinical Trials	Random Sequence Generation	Allocation Concealment	Blinding of Participants, Personnel	Blinding of Outcome Assessment	Incomplete Outcome Data		Selective Reporting	Other Bias
Comert et al [72]	?	?	-	+	+		+	+
Del Fabbro et al [73]	+	?	?	−	+		+	+
Gassling et al [74]	?	+	?	+	+		+	+
Gurler et al [75]	+	?	-	+	−	−	+	+
Nizam et al [41]	+	?	-	+	?		+	+
Olgun et al [45]	+	?	-	?	+		+	+
Pichotano et al [42]	+	+	+	+	+		+	+
Tatullo et al [76]	+	?	−	?	+		+	+
Zhang et al [77]	?	?	-	?	+		+	+

+: Low;?: Unclear;−: High.

**Table 4 materials-13-00622-t004:** Summary of outcome measures of studies reviews for PRF on sinus augmentation.

Author, Year Country	Outcome Measures			
Bolukbasi et al, 2015 Turkey [69]	**Radiographic assessment **(T_0_ − T_5_) (mean ± standard deviation) Relationship between sinus-graft height and the implant: BL/IL ratio ^a^: Intervention group: 1.36 ± 0.04 // Control group: 1.36 ± 0.03 During study period, intervention group showed less change in BL/IL values than control group (p = 0.022) Change in the height of grafted sinus: GSH/OSH ratio ^b^: Intervention group: 4.49 ± 0.29 // Control group: 4.14 ± 0.20 (p = 0.093) **Histomorphometric assessment (%)** (mean ± standard deviation) New bone formation: Intervention group: 35 ± 8.6 // Control group: 32.9 ± 9.7 (p = 0.61) Connective tissue: Intervention group: 30.6 ± 7.5 // Control group: 33.9 ± 9.2 (p = 0.34) Biomaterial remnants: Intervention group: 33.1 ± 6.3 // Control group: 33.8 ± 8.6 (p = 0.87)			
Bosshardt et al, 2014 Switzerland [71]	**Histomorphometric assessment (%)** (mean ± standard deviation) New bone formation: Intervention group (PRF): 28.6 ± 6.9 // Control group: 28.7 ± 5.4 Residual bone substitute material: Intervention group (PRF): 25.6 ± 8.7 // Control group: 25.5 ± 7.6 Soft Tissue: Intervention group (PRF): 45.7 ± 9.3 // Control group: 45.8 ± 3.2 No statistically significant differences			
Choukroun et al, 2006 France [70]	**Histomorphometric assessment (%)** New bone formation: FDBA + PRF after 4 months: 65% (mean: 20.9% range: 18.6–30.3%) FDBA after 8 months: 69% (mean: 20.3% range: 18–23.7%)			
Comert et al, 2017 Turkey [72]	**Histomorphometric assessment (%)** (mean ± standard deviation) New bone formation: Group A (P-PRP + β-TCP): 34.83 ± 10.1 // Group B (PRF + β-TCP): 32.03 ± 6.3 // Control group (β-TCP): 33.40 ± 10.4 (p = 0.83) Residual graft particle area: Group A (P-PRP + β-TCP): 28.98 ± 7.9 // Group B (PRF + β-TCP): 32.66 ± 7.5 // Control group (β-TCP): 33.39 ± 10.3 (p = 0.69) Soft (fibrous) tissue area: Group A (P-PRP + β-TCP): 36.19 ± 13.9 // Group B (PRF + β-TCP): 35.31 ± 10.8 // Control group (β-TCP): 36.21 ± 10.6 (p = 0.98)			
Del Fabbro et al, 2013 Italy [73]	**Quality of life in the postoperative treatment** Pain (VAS Score): The use of PRGF resulted in significant reduction of the perceived pain during the second and third postoperative day as compared to the control group. From day 4, the mean VAS scores of the two groups were similar. Patients in the PRGF group reported consistently less swelling, less hematoma and less discomfort regarding chewing and speaking throughout the evaluation period. Bleeding was significantly lower in the first 2 days in the PRGF group (p = 0.01)			
Gassling et al, 2013 Germany [74]	**Histomorphometric assessment (%)** New (vital) bone formation: Intervention group (PRF): mean: 17.0% range: 7.8–27.8% // Control group (collagen membrane): mean: 17.2% range: 8.5–24.2% Residual bone-substitute material: Intervention group (PRF): mean: 15.9% range: 0.9–33.4% // Control group (collagen membrane): mean: 17.3% range: 0.7–33.5%			
Gurler et al, 2016 Turkey [75]	**Postoperative complications of direct sinus lifting** Gradual improvements in the L-PRF group regarding postoperative pain, swelling, sleeping, eating, phonetics, activities of daily living and missed work days, but not statistically significant (p > 0.05) Wound healing in both groups was uneventful Healing index scores of the L-PRF group were higher than the control group on the 7^th^ (4.2 ± 0.9 vs. 3.6 ± 0.7) and 14^th^ (4.7 ± 0.4 vs. 4.4 ± 0.5) days, but not statistically significant (p = 0.13, p = 0.19; respectively)			
Nizam et al, 2017 Turkey [41]	**Histomorphometric assessment (%)** (mean ± standard deviation) New bone formation: Intervention group (L-PRF): 21.4 ± 8.8 // Control group: 21.2 ± 5.6 (p = 0.96) Residual bone graft: Intervention group (L-PRF): 25.9 ± 9.5 // Control group: 32.8 ± 5.9 (p = 0.06) Bone graft in contact with newly formed bone: Intervention group (L-PRF): 47.3 ± 12.3 // Control group: 54.0 ± 8.4 (p = 0.16) Soft tissue component: Intervention group (L-PRF): 52.7 ± 12.5 // Control group: 45.9 ± 8.4 (p = 0.16) **Radiographic assessment (mm)** (mean ± standard deviation) Augmented bone height: Intervention group (L-PRF): 13.6 ± 1.1 // Control group: 13.5 ± 1.2 (p = 0.88) **Implant survival (after 1 year of follow-up)** 100% in both groups			
Olgun et al, 2018 Turkey [45]	**Histomorphometric assessment (%)** (mean ± standard deviation) New bone formation: Intervention group (T-PRF): 16.6 ± 1.1 // Control group: 17.3 ± 2.5 (p = 0.85) Cancellous bone ratio: Intervention group (T-PRF): 24.0 ± 1.5 // Control group: 22.7 ± 2.6 (p = 0.74) **Radiographic assessment** (mean ± standard deviation or median (IQR)) Bone height (mm): Intervention group (T-PRF): 11.7 ± 2.4 // Control group: 19.9 ± 7.4 (p = 0.05) Bone volume (mm^3^): Intervention group (T-PRF): 172.7 (82.6) // Control group: 264.6 (70.2) (p = 0.001) Bone density (hu): Intervention group (T-PRF): 86.7 (43.6) // Control group: 160.8 (63.6) (p = 0.001) Control group had 53% better volume, 86% better density and 69% better height compared to T-PRF group. **Clinical assessment** (mean ± standard deviation) Implant stability (Implant stability quotient): Intervention group (T-PRF): 68.5 ± 8.9 // Control group: 66.4 ± 8.3 (p = 0.61)			
Pichotano et al, 2018 Brazil [42]	**Radiographic assessment** Graft volume dimensional changes (mean graft reduction between T_1_ − T_2_^c^) (mean ± standard deviation) Cm^3^= Intervention group (L-PRF): 0.58 ± 0.26 // Control group: 0.55 ± 0.34 (p = 0.78) % = Intervention group (L-PRF): 33.1 ± 10.7 // Control group: 36.7 ± 15.8 (p = 0.47) **Histomorphometric assessment (%)** (mean ± standard deviation) New bone formation: Intervention group (L-PRF): 44.6 ± 13.9 // Control group: 30.0 ± 8.4 (p = 0.008) Residual graft material: Intervention group (L-PRF): 3.6 ± 4.2 // Control group: 13.7 ± 9.9 (p = 0.01) Soft (fibrous) tissue: Intervention group (L-PRF): 26.6 ± 11.1 // Control group: 30.6 ± 12.5 (p = 0.38) **Clinical assessment** Implant stability (Implant stability quotient) (mean ± standard deviation) After implant placement: Intervention group (L-PRF): 60.9 ± 9.3 // Control group: 75.1 ± 5.7 (p < 0.001) At implant loading: Intervention group (L-PRF): 76.1 ± 5.9 // Control group: 75.7 ± 6.1 (p = 0.99) Implant survival (after 1 year of follow-up): 100% in both groups			
Tatullo et al, 2012 Italy [76]	**Histomorphometric assessment (%)**			
		Early protocol	Intermediate protocol	Late protocol
	Medullary spaces	I (PRF): 70.2 // C: 68.4	I (PRF): 70.0 // C: 68.2	I (PRF): 61.4 // C: 58.2
	Osteoid borders	I (PRF): 7.01 // C: 5.12	I (PRF): 3.84 // C: 3.12	I (PRF): 3.5 // C: 2.9
	Trabecular bone	I (PRF): 22.8 // C: 26.4	I (PRF): 26.2 // C: 28.7	I (PRF): 37.1 // C: 38.9
	**Clinical assessment** Implant stability (Implant stability quotient): No statistically significant differences were found between groups in each of the protocols performed Implant survival (after 36 ± 10 months of follow-up): 100% in both groups			
Zhang et al, 2012 China [77]	**Histomorphometric assessment (%)** (mean ± standard deviation) New bone formation: Intervention group (PRF): 18.4 ± 5.6 // Control group: 12.9 ± 5.3 (p = 0.14) Residual bone substitute (BioOss): Intervention group (PRF): 19.2 ± 6.9 // Control group: 28.5 ± 12.0 (p = 0.14) Bone-to-bone substitute contact: Intervention group (PRF): 21.5 ± 14.6 // Control group: 18.6 ± 5.4 (p > 0.05)			

Abbreviations: T_0_: 10 days after sinus lifting surgery; T_5_: 24 months after prosthetic loading; BL: distance from the top of the bone-to-implant contact region to the head of the fixture (bone level); IL: distance from the apex to the head of the fixture (implant length); GSH distance from the intraoral marginal bone to the grafted sinus floor above the lowest part of the original sinus height (grafted sinus height); OSH: distance from the intraoral marginal bone to the lowest point of the original sinus floor (original sinus height); FDBA: freeze-dried bone allograft; PRF: Plasma-Rich Fibrin; P-PRP: Platelet-Rich Plasma; β- TCP: Beta-Tricalcium Phosphate; L-PRF: Leukocyte-PRF; T-PRF: Titanium-PRF; IQR: Interquartile range; VAS: Visual Analogue Scale; PRGF: Plasma Rich in Growth Factors; I: Intervention group; C: Control group; ^a^ BL/IL ratio: shows the mean change in height of grafted sinus floor for implant (values ≥ 1 indicates that the grafted sinus covers the implant apex); ^b^ GSH/OSH ratio: shows the mean change in height of grafted sinus (values ≥ 1 shows that the grafted sinus floor is above the original sinus height); ^c^ T_1_: mean graft volume immediately after sinus augmentation, T_2_: mean graft volume after 4 months for intervention group and 8 months for control group.

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
