# Peer review of "Platelet-Rich Plasma in Maxillary Sinus Augmentation: Systematic Review"

_materials, 2020, doi:10.3390/ma13030622_

Round 1

Reviewer 1 Report

In this review, the authors attempt to consolidate within MEDLINE the importance of using PRF along with biomaterials in sinus floor augmentation to evaluate if this may be significant in the improving the augmentation process. After a selection process made by the authors, only 11 studies on the sole use of platelet concentrates on maxillary sinus augmentation had been considered viable towards the discussion due to various reasons. A further 21 academic publications on PRF+ biomaterials as a comparative measure.  

This review reads more like a laboratory record book rather than an authoritative academic discussion. There is a lack of an illustration at the onset of this manuscript to help quickly to the reader what this review entices. Furthermore, there is a major absence of flow in the writing due to the truncation of the sentences. For instance, “All articles were in English” is often used in the start of sections but is it really necessary? Every individual section reads more like a report rather than a discussion of sorts. The presentation of data is relatively bland and dry.

In brief, this review is very bland in the reviewer opinion and is only well suited for more technical journal for niche researchers rather than in MDPI materials. Furthermore, there is very little conclusions that can be drawn at the end of this manuscript in view of the narrow scope of the literature search.

Author Response

Reply

Thank you. We have considered your comments and improved the manuscript according to the observations mentioned. Changes are highlighted in the revised manuscript. We have discussed the findings of our review and gave our opinion regarding the use of platelet concentrates in sinus lift procedures considering the available evidence and the potential advantages of using PRF as a grafting biomaterial. Unfortunately, no firm conclusions can be drawn. As we mention in the review, the clinical applications of PRF in dentistry and, specifically, in sinus lift procedures are relatively new, proven by the scarce available literature. For that reason we decided to conduct a comprehensive review and a wide search to retrieve most of published studies that answered our focus PICO question. Moreover, studies were not of high methodological quality and were heterogeneous in terms of intervention, comparison and outcome measure precluding us to provide with recommendations for the clinical practice. That is a limitation that we account for.

Reviewer 2 Report

This is an interesting review of the literature on the possible use of platelet-rich plasma in maxillary sinus augmentation.

First of all, I intend to congratulate the authors both for the choice of the scientific topic and for the precision in carrying out the revision.

Some criticisms are present:

-Abstract: this section must be completely rewritten.

First of all it is necessary to insert some initial sentences on the possible effect of platelet rich plasma in the oral healing processes; then all the central part relating to the search engines (PUBMED; etc) must be removed as it is not useful in this section. I would focus the reader's attention more on the general problem.

-Keywords: remove the words "dental implants", "systematic review" and "meta-analyzes"

-Rationale: in this section a part relating to the other uses in the dental field of platelets should be added, in the broader discussion of the various replacement techniques.

In this regard, I recommend inserting the following security work in the reference section and in this part.

Chieruzzi M, Pagano S, Moretti S, Pinna R, Milia E, Torre L, Eramo S.

Nanomaterials for Tissue Engineering In Dentistry. Nanomaterials (Basel). 2016

Jul 21;6(7). pii: E134. doi: 10.3390/nano6070134. Review. PubMed PMID: 28335262;

PubMed Central PMCID: PMC5224610.

-Section 3.5 Remove the initials of the authors' names as they are not functional to the work

-Section 4.2 Remove the expression "all included studies were in English"; this has already been stated in the study selection criteria

In the discussions section, some statements should be added on what is expressed in the literature about the advantages and disadvantages of using protein-rich plasma in surgical procedures.

Author Response

This is an interesting review of the literature on the possible use of platelet-rich plasma in maxillary sinus augmentation.

First of all, I intend to congratulate the authors both for the choice of the scientific topic and for the precision in carrying out the revision.

Some criticisms are present:

-Abstract: this section must be completely rewritten.

First of all it is necessary to insert some initial sentences on the possible effect of platelet rich plasma in the oral healing processes; then all the central part relating to the search engines (PUBMED; etc) must be removed as it is not useful in this section. I would focus the reader's attention more on the general problem.

Reply.

Thank you. We have considered your recommendations and rewritten the abstract as suggested.

-Keywords: remove the words "dental implants", "systematic review" and "meta-analyzes"

Reply.

Thank you. We removed those keywords from the revised manuscript.

-Rationale: in this section a part relating to the other uses in the dental field of platelets should be added, in the broader discussion of the various replacement techniques.

 In this regard, I recommend inserting the following security work in the reference section and in this part.

Chieruzzi M, Pagano S, Moretti S, Pinna R, Milia E, Torre L, Eramo S.

Nanomaterials for Tissue Engineering In Dentistry. Nanomaterials (Basel). 2016

Jul 21;6(7). pii: E134. doi: 10.3390/nano6070134. Review. PubMed PMID: 28335262;

PubMed Central PMCID: PMC5224610.

Reply.

Thank you. We described in the Rationale section the main uses of platelet concentrates in the dental field, taken as reference the article suggested. The paragraph is highlighted in the revised manuscript with corresponding reference number 38.

 -Section 3.5 Remove the initials of the authors' names as they are not functional to the work

Reply.

Thank you. We removed the initials of the authors name as recommended.

-Section 4.2 Remove the expression "all included studies were in English"; this has already been stated in the study selection criteria

Reply.

Thank you. We removed that expression from section 4.2.

In the discussions section, some statements should be added on what is expressed in the literature about the advantages and disadvantages of using protein-rich plasma in surgical procedures.

Reply.

Thank you. We added statements regarding the advantages and disadvantages of using PRF in surgical procedures such as sinus lift augmentation and highlighted in the revised manuscript.

Reviewer 3 Report

Platelet-rich plasma in maxillary sinus augmentation: Systematic review is very interesting paper with high importance in medicine and science of materials. Some improvements shall be finished in this text.

Task 5.3 is missing (Innovation and challenges of this systematic study)

One sentence (proposal) shall be added in conclusion as future work to find some solution for limitations in this work

Title of Table 1 and Table 2 to start: Descriptive characteristics ... (without "of")

Title of Table 3 to start: Bias of included randomized.. (without "of")

Text in Figure 1 A and 1 B shall be written in a decreased font size in order to see all numbers such as 18, 11 132

Author Response

Platelet-rich plasma in maxillary sinus augmentation: Systematic review is very interesting paper with high importance in medicine and science of materials. Some improvements shall be finished in this text.

Task 5.3 is missing (Innovation and challenges of this systematic study)

Reply

Thank you. We have added the Innovation and challenges in the discussion section as task 5.3 and highlighted it in the revised manuscript.

One sentence (proposal) shall be added in conclusion as future work to find some solution for limitations in this work

Reply

Thank you. We have added a one sentence proposal in the conclusion section for future work: “Nevertheless, well-designed and conducted RCTs with the same PRF protocol preparation, blinding of personnel in charge of outcome assessment and long follow-up periods are necessary to confirm the available results, determine their clinical implications and recommendations for the clinical practice.”

Title of Table 1 and Table 2 to start: Descriptive characteristics ... (without "of")

--Reply

Thank you. We have corrected the title of Tables 1 and 2 as suggested, and highlighted it in the manuscript.

Title of Table 3 to start: Bias of included randomized. (without "of")

--Reply

Thank you. We have corrected the title of Table 3 and highlighted it in the manuscript

Text in Figure 1 A and 1 B shall be written in a decreased font size in order to see all numbers such as 18, 11 132

--Reply

Thank you. We have made the corrections and written the text of Figures 1A and 1B in a smaller font size.

Reviewer 4 Report

This is a timely and interesting literature review on the impact of platelet concentrates on the outcome of sinus lifts (in the presence and absence of various grafting materials). This review will be of relevance to clinicians and scientists working in dentistry and maxillofacial surgery; although, it does not seem to fall within the general purview of this journal.

The search criteria and meta-analysis are rigorous and comprehensively described; and the conclusions are supported by the evidence included in the review. The principal conclusion being that the application of the patient’s autologous platelet-enriched plasma is a relatively new approach (~20 years) in maxillofacial surgery and more RCTs are required in order for firm clinical recommendations to be drawn.

The manuscript is generally comprehensible; although, many grammatical anomalies (also typing and formatting errors) need to be corrected prior to publication.

Throughout the manuscript, the authors should ensure that all abbreviations are defined prior to use.

The PICO tool should be referenced when it is mentioned on page 3.

I have no confidential comments for the editors.

Author Response

This is a timely and interesting literature review on the impact of platelet concentrates on the outcome of sinus lifts (in the presence and absence of various grafting materials). This review will be of relevance to clinicians and scientists working in dentistry and maxillofacial surgery; although, it does not seem to fall within the general purview of this journal.

The search criteria and meta-analysis are rigorous and comprehensively described; and the conclusions are supported by the evidence included in the review. The principal conclusion being that the application of the patient’s autologous platelet-enriched plasma is a relatively new approach (~20 years) in maxillofacial surgery and more RCTs are required in order for firm clinical recommendations to be drawn.

The manuscript is generally comprehensible; although, many grammatical anomalies (also typing and formatting errors) need to be corrected prior to publication.

--Reply

Thank you. An English colleague has revised the manuscript and corrected the grammatical, typing and formatting errors. Changes have been highlighted in the revised manuscript.

Throughout the manuscript, the authors should ensure that all abbreviations are defined prior to use.

--Reply

Thank you. We have revised the manuscript and ensured that all abbreviations were defined prior to use. Some of the abbreviations were not previously defined, so we have corrected this issue and highlighted them in the revised manuscript.

The PICO tool should be referenced when it is mentioned on page 3.

--Reply

Thank you. We have added the corresponding reference for the PICO tool:

Shamseer, L; Moher, D; Clarke, M; Ghersi, D; Liberati, A; Petticrew, M, Shekelle, P; Stewart, L; the PRISMA-P Group. Preferred reporting items for systematic review and meta-analysis protocols (PRISMA-P) 2015: elaboration and explanation. BMJ. 2014; 349:g7647. doi: 10.1136/bmj.g7647

I have no confidential comments for the editors.